# Mechanical Characterization of Dissolving Microneedles: Factors Affecting Physical Strength of Needles

**DOI:** 10.3390/pharmaceutics16020200

**Published:** 2024-01-30

**Authors:** Daisuke Ando, Megumi Miyatsuji, Hideyuki Sakoda, Eiichi Yamamoto, Tamaki Miyazaki, Tatsuo Koide, Yoji Sato, Ken-ichi Izutsu

**Affiliations:** 1Division of Drugs, National Institute of Health Sciences, 3-25-26 Tonomachi, Kawasaki-ku, Kawasaki 210-9501, Kanagawa, Japaneyamamoto@nihs.go.jp (E.Y.); koide@nihs.go.jp (T.K.); kenichi-izutsu@iuhw.ac.jp (K.-i.I.); 2Division of Medical Devices, National Institute of Health Sciences, 3-25-26 Tonomachi, Kawasaki-ku, Kawasaki 210-9501, Kanagawa, Japan; sakoda@nihs.go.jp; 3Department of Pharmaceutical Sciences, School of Pharmacy, International University of Health and Welfare, 2600-1 Kitakanemaru, Ohtawara 324-8501, Tochigi, Japan

**Keywords:** dissolving microneedles, transdermal drug delivery system, mechanical characterization, quality control, fracture force

## Abstract

Dissolving microneedles (MNs) are novel transdermal drug delivery systems that can be painlessly self-administered. This study investigated the effects of experimental conditions on the mechanical characterization of dissolving MNs for quality evaluation. Micromolding was used to fabricate polyvinyl alcohol (PVA)-based dissolving MN patches with eight different cone-shaped geometries. Axial force mechanical characterization test conditions, in terms of compression speed and the number of compression needles per test, significantly affected the needle fracture force of dissolving MNs. Characterization using selected test conditions clearly showed differences in the needle fracture force of dissolving MNs prepared under various conditions. PVA-based MNs were divided into two groups that showed buckling and unbuckling deformation, which occurred at aspect ratios (needle height/base diameter) of 2.8 and 1.8, respectively. The needle fracture force of PVA-based MNs was negatively correlated with an increase in the needle’s aspect ratio. Higher residual water or higher loading of lidocaine hydrochloride significantly decreased the needle fracture force. Therefore, setting appropriate methods and parameters for characterizing the mechanical properties of dissolving MNs should contribute to the development and supply of appropriate products.

## 1. Introduction

Microneedles (MNs), a novel transdermal drug delivery system, are a promising alternative to conventional injections. MNs mainly consist of 10–1000 needles that are smaller than 1 mm in height. They disrupt the skin’s stratum corneum barrier without pain, enhance the permeation of topically applied drugs, and can be self-administered [1]. MNs are classified into four categories according to their structure and mechanism of action: (1) solid MNs for skin pretreatment to enhance permeability; (2) coated MNs with drug coating that dissolves in the skin; (3) hollow MNs for drug solution injection; and (4) dissolving MNs that encapsulate the drug, dissolve in the skin, and release the drug [2,3]. Each type of MN has its own advantages and limitations. The first three types are mainly made from non-biodegradable materials, like silicon or metal, which offer good mechanical strength. However, there is a potential risk of residues of broken needles in the skin. Currently, dissolving MNs that are constituted from water-soluble polymers such as sodium hyaluronate [4], polyvinyl pyrrolidone [5], and polyvinyl alcohol (PVA) [6] are regarded as the safest, as they dissolve completely in the skin. Dissolving MNs can also enhance the chemical stability of encapsulated drugs and vaccine antigens, and hence do not require cold storage [7,8]. They do not generate bio-medical waste and cannot be reused [9], which are major advantages. Owing to this, dissolving MNs could enhance the accessibility of essential medicines and vaccines in low- and middle-income countries [10].

There are no currently approved dissolving MNs for medical use by the U.S. Food and Drug Administration and the Pharmaceuticals and Medical Devices Agency of Japan [11]. A major hurdle in MN development is the lack of quality evaluation methods for this novel dosage form [12,13,14]. Unlike conventional pharmaceuticals, quality assurance of MNs cannot be achieved by referring to published guidelines and the expertise of pharmaceutical companies. The absence of guidelines places a burden on applicants, such as establishing evaluation methods and confirming validity, which is unnecessary for other dosage forms, thereby hindering product development. Recently, a Regulatory Working Group (RWG) was established to collaborate with industry, government, and academia to identify critical quality attributes (CQAs) and standardize test methods for MNs [15,16]. The RWG proposed that the delivered dose, puncture performance, dissolution, physical stability, mechanical strength, and needle morphology were high-priority CQAs [17].

MNs must be strictly designed with a mechanical strength greater than the force required to penetrate the skin for safe and effective transdermal drug delivery. Axial force mechanical characterization is the most frequently used test for evaluating the mechanical properties of MNs [13,18,19,20,21,22,23]. In this test, MN patches are subjected to a force perpendicular to their base plate. The MN patches are pressed by a sensor probe at a defined speed and a mechanical test station records both force and displacement. The force–displacement curves show a sudden drop in force upon needle fracture, and in general, the maximum force applied immediately before dropping is termed the needle fracture force [18,20,21]. This mode of failure is known as buckling deformation. MNs with a high aspect ratio (needle height/base diameter) mainly show buckling deformation even though the stress at the point of failure is less than the maximum stress that the material can withstand.

Many dissolving MNs composed of water-soluble polymers have weak mechanical strength due to their lower Young’s modulus and tensile strength compared to conventional MNs made of metals and silicon [24]. It is also expected that the mechanical strength of dissolving MN will be affected by the embedded drug or vaccine antigen. Therefore, it is important to properly characterize the mechanical properties of dissolving MNs. However, the importance of MN mechanical characterization test protocols is underestimated. To this end, previous studies have performed mechanical characterization tests on dissolving MNs with different geometries under different test conditions; hence, it is difficult to directly compare the results. In this study, we investigated the effects of test parameters on the measured value by systemically varying them. To achieve this, dissolving MN patches with eight different cone-shaped geometries were prepared by the micromolding method using PVA as the base material because of its water solubility, biocompatibility, and mechanical strength. We investigated the effects of axial force mechanical characterization test conditions in terms of compression speed and number of compression needles per test on the needle fracture force. After adjusting for test conditions, we examined differences in the needle fracture force of PVA-based dissolving MNs prepared under various conditions on the basis of the needle aspect ratio, water content, and drug content. The findings of this study can provide appropriate parameters for the experimental setup and optimization of dissolving MNs.

## 2. Materials and Methods

### 2.1. Materials

PVA with an average molecular weight of between 9000 and 10,000 was purchased from Sigma Aldrich Inc. (St. Louis, MO, USA). Lidocaine hydrochloride (LID) was purchased from MP Biomedicals (Santa Ana, CA, USA). Aluminum MN master molds were purchased from Tokai Azumi Techno (Tsu, Japan).

### 2.2. Fabrication of Dissolving Microneedles

Micromolding was used to fabricate dissolving MNs with PVA as previously described [25,26,27]. Different MN master molds were fabricated as shown in Table 1. Inverse replicates of the master molds were prepared using polydimethylsiloxane (PDMS) purchased from Dow Corning (Midland, MI, USA). Approximately 1.0 g of 30% (*w*/*w*) PVA aqueous solution was poured into 2.1 × 2.1 cm PDMS molds and centrifuged at 2380× *g* for 15 min. The molds were dried in an oven at 35 °C with dry silica gel for 24 h. The baseplate was reinforced with a polyethylene terephthalate film. Finally, demolded samples were cut into circular shapes with 12 mm diameters and stored in aluminum plastic-laminated packaging with dry silica gel at 25 °C.

### 2.3. Scanning Electron Microscopy (SEM)

The MN patches were mounted on 45°-angled aluminum stubs using double-sided carbon adhesive tape (Nisshin EM, Tokyo, Japan) and imaged using SEM (TM3030Plus, Hitachi High-Tech, Tokyo, Japan) at 50× magnification.

### 2.4. Micro X-ray Computed Tomography (Micro-CT)

Micro-CT (nano3DX; Rigaku, Tokyo, Japan) equipped with a lens (field of view: 5.32 × 5.32 mm) was used to scan the region of interest. The radiation source was CuKα (40 kV, 30 mA). A total of 400 projections were obtained with a binning of 1 and an exposure time of 10.8 s/projection, allowing for an achievable voxel size of 2.6 µm. Reconstruction software (CT Reconstruction App version 2.1.0.0; Rigaku) and image analysis software (Image J version 1.50i; National Institute of Health, Bethesda, MD, USA) were used to construct the tomograms and measure the actual dimensions of dissolving MNs.

### 2.5. MN Mechanical Characterization Test

The mechanical strength of the MNs was evaluated based on axial force mechanical characterization using a digital force gauge (ZTA-50N) equipped with a motorized test stand (EMX-1000N) from Imada (Toyohashi, Japan). As shown in Appendix A, the MN patch was fixed to the platform with double-sided tape and pressed by a sensor probe equipped with a rod diameter of 2, 3, 4, 5, or 13.3 mm (PG-2, PG-3, PG-4, PG-5, or A-2, respectively; Imada). The compression speed ranged from 0.5 to 100 mm/min. Force–displacement curves were recorded continuously from 0.050 N until a sudden drop in force was observed or up to 32 N. As shown in Appendix A, we interpreted the peak value (local maximal force) as the needle fracture force. The fracture force per needle was calculated by dividing the measured fracture force by the number of needles that were fractured. After compression, the MN shape was determined using optical microscopy (RX-100; Hirox, Tokyo, Japan). We used dissolving MNs that had been stored over 2 days for these experiments, except for assessing the effect of water content.

### 2.6. Quantification of Water Content

The water content of the MN patch was assessed using a Karl Fischer titration (CA-200; Mitsubishi Chemical Co., Tokyo, Japan). The reagents used were Aquamicron AX as the anolyte and Aquamicron CXU as the catholyte. A pre-weighted amount of an MN patch using an XPR56DRV electronic scale (Mettler Toledo, Tokyo, Japan) was dispersed into an anolyte solution in the reaction vessel. The precision, which is the relative standard deviation of 0.5% water content with the sensitivity, ranged from 10 µg to 100 mg.

### 2.7. Quantification of Drug Content

The LID content was quantified using high-performance liquid chromatography (HPLC; LC20A) with UV detection (SPD-20A; Shimadzu, Kyoto, Japan). Needles were removed from the baseplate using a razor and collected per patch. They were dissolved in 10 mL phosphate-buffered saline (pH 7.4). HPLC was performed using a Shim-pack GIS C18 column (4.0 × 150 mm, 5 µm) at an injection volume of 50 µL. The mobile phase consisted of 0.1% (*v*/*v*) trifluoroacetic acid in acetonitrile water (20:80, *v*/*v*). The column temperature and flow rate were 40 °C and 1.0 mL/min, respectively. The detection wavelength was 262 nm. LID was eluted at 5 min. The peak area of LID was used for quantification of unknown samples. The standard curve was linear between 1.25 and 20 µg/mL, with a correlation coefficient of 1.000.

### 2.8. Statistical Analysis

The results are presented as mean ± standard deviation (SD). Microsoft Excel version 2016 was used to estimate the slope and interception of the curves by least squares regression. GraphPad Prism software version 8.4.3 (GraphPad, Inc. San Diego, CA, USA) was used for statistical analysis. One-way ANOVA with Tukey’s post hoc test and Student’s *t*-test were used for comparison, and statistically significant differences between the test groups were set at a *p*-value of less than 0.05.

## 3. Results

### 3.1. Fabrication of Dissolving MNs

Our recent study demonstrated that a sufficient needle height (>600 μm) is required to overcome the viscoelasticity of the skin and achieve effective transdermal drug delivery [26]. Hence, we designed eight different geometries of the cone-shaped MNs with needles higher than 600 µm. The MN design dimensions, such as needle height, needle base and tip diameters, interspacing of needles at the tip, and number of needles per patch, are shown in Table 1.

Figure 1 shows the SEM images depicting the PVA-based MNs. These results suggest that PVA-based MNs produced by micromolding were uniform in size with sharp tips. The actual dimensions of the PVA-based MNs were measured using micro-CT, as shown in Table 2. The resultant average needle height and interspacing of the needles were less by approximately 5% than the MN design dimensions because dissolving MNs reduce in size due to the drying process. These shrinkages have been previously reported [28]; therefore, it is important to measure the actual dimensions of the dissolving MNs. Fabrication methods, such as polymer concentration and viscosity, as well as temperature and drying time in the oven, may affect the shrinkage of dissolving MNs [29]. Establishing both the fabrication and evaluation methods should be required for the product optimization and their continuous supply. Overall, the actual dimensions were in good agreement with the design dimensions; thus, our dissolving MNs could be used for research purposes.

### 3.2. Effect of Compression Speed on the MN Mechanical Characterization Tests

Previously, the compression speeds were adjusted from 0.6 to 66 mm/min [18,21,22,30,31,32,33,34,35,36,37]; however, the effect of these speeds on the test results was not examined. Hence, we determined the influence of the compression speed on mechanical characterization test results.

To confirm the deformation modes of our PVA-based MNs, Model 2 MNs (11 × 11 array, base diameter of 179 µm, and height of 573 µm), which resemble MNs reported by Hiraishi [21], were examined at a compression speed of 0.5 mm/min using a 5 mm diameter rod. Hiraishi et al. [21] demonstrated that needle buckling deformation was identified using sodium hyaluronate-based MNs that have a high aspect ratio (800 µm in height and less than 200 µm in base diameter). As shown in Appendix A, a local maximal force was clearly detected from the force–displacement curve. Appendix A shows that the needle tips bent at 90 degrees after compression. These results demonstrate that Model 2 MNs buckled during the test and the needle fracture force could be evaluated. Hence, Model 2 MNs were preferred for this experimental setup.

Figure 2 shows the effect of the compression speed ranging from 0.5 mm/min to 100 mm/min on the needle fracture force. The measured fracture force increased in a compression speed-dependent manner. The needle fracture force at the minimum (0.5 mm/min) and maximum (100 mm/min) speed were 0.145 ± 0.006 and 0.203 ± 0.015 N/needle, respectively, showing a significant 1.4-fold difference (Student’s *t*-test, *p* < 0.05). The number of fracture needles was strictly controlled (60.8 ± 1.0 to 63.0 ± 0.0 needles per test) at different compression speeds. In addition, the relative standard deviations of the needle fracture force were controlled from 3.8 to 8.8% at different compression speeds, showing good reproducibility at an individual compression speed. These results suggest that directly comparing needle fracture force at different compression speeds is not feasible. For quality control testing, it is essential to evaluate the needle fracture force at the same compression speed.

### 3.3. Effect of Number of Needles Compressed on MN Mechanical Characterization Tests

The number of compressed needles differs among studies as it depends on the MN patch design, which ranges from approximately 10 to 1000 needles, and compression rod diameter [18,23,36,38,39]. However, their effect on the test results has not been previously examined. We determined the influence of the number of needles on the MN mechanical characterization test.

We evaluated PVA-based MNs with the same needle shape but different numbers of needles per patch (7 × 7, 11 × 11, and 16 × 16 array), namely Models 7, 2, and 8, respectively. Using a 5 mm diameter rod and a compression speed of 0.5 mm/min, the number of compressed needles was 20.3 ± 1.0, 62.0 ± 1.2, and 131.5 ± 1.0, respectively. As shown in Figure 3A, the needle fracture force decreased as the number of needles on a patch increased. The 16 × 16 array needle fracture force (0.134 ± 0.009 N/needle) was significantly lower than that of the 7 × 7 array (0.182 ± 0.019 N/needle) (one-way ANOVA, *p* < 0.05).

Figure 3A denotes the evaluation results of different MN models with different needle spacing (the effect of different designs could not be ruled out). We further elucidated the effect of needle number on MNs using rods of different diameters for a single model (Model 2) (Figure 3B). By changing the rod diameter to 2, 3, 4, 5, and 13.3 mm, the number of compressed needles were 9.0 ± 0.0, 21.0 ± 0.0, 38.3 ± 1.9, 62.8 ± 0.5, and 121.0 ± 0.0, respectively. Figure 3B shows that the needle fracture force decreased as the number of needles increased per test, which is in good agreement with Figure 3A. The needle fracture force at the minimum number of needles (9.0 needles/test) and the maximum number (121 needles/test) were 0.192 ± 0.012 and 0.107 ± 0.015 N/needle, respectively, showing a significant 0.56-fold difference (Student’s *t*-test, *p* < 0.05). These results suggest that the number of compressed needles strongly affects the needle fracture force.

### 3.4. Effect of Needle Dimensions on Mechanical Strength of Dissolving MNs

To determine the effect of the MN dimensions on the mechanical properties, we compared the needle fracture force on the basis of needle height and base diameter. We used a constant test condition under a compression rate of 0.5 mm/min using a 5 mm diameter rod for subsequent experiments. Figure 4 shows the force–displacement curves for PVA-based Model MNs 4, 5, and 6 with different actual needle lengths of 570, 851, and 1133 µm, respectively. Each MN had a fixed design base diameter of 300 µm. As shown in Figure 4B,C, MNs with heights of 851 and 1133 µm showed local maximum points, demonstrating buckling deformation. On the other hand, those with heights of 570 µm did not show these points (Figure 4A). MNs with heights of 570 µm showed a progressive deformation of the needles, starting near the tip and moving downward with increasing force, but never showed a catastrophic buckling event at a single point of failure. These results suggest that dissolving MNs can be divided into two groups that show buckling and unbuckling deformation on the basis of their aspect ratio. In the present study, buckling and unbuckling occurred at an aspect ratio of 2.8 and 1.8, respectively.

Figure 5A shows that the needle fracture force decreased with increasing actual needle height from 0.253 ± 0.011 N/needle for 851 µm to 0.135 ± 0.015 N/needle for 1133 µm with a fixed design base diameter of 300 µm. Figure 5B shows that the needle fracture force of PVA-based Models 1, 2, and 3 with 139, 179, and 216 µm actual base diameters with a fixed design needle height of 600 µm were 0.083 ± 0.006, 0.136 ± 0.015, and 0.203 ± 0.045 N/needle, respectively. There was a negative correlation between the needle’s aspect ratio and needle fracture force of dissolving MNs (Figure 5C). These results are consistent with the behavior of solid MNs designed from poly-lactic-co-glycolic acid (PLGA) [18,40]. These results concurred with Euler’s formula, which postulates that the needle fracture force enhances with a decreasing needle aspect ratio since the critical buckling (lateral deflection) load of a column decreases with an increasing column aspect ratio [41]. For both design and quality control of dissolving MNs, we presume that it is necessary to set specifications for not only the needle height but also the needle aspect ratio. 

### 3.5. Effect of Physicochemical Properties on the Mechanical Strength of Dissolving MNs

Considering the fabrication method of dissolving MNs, the residual water content may affect their mechanical strength. Figure 6A depicts the water content in PVA-based Model 2 MNs determined by the Karl Fisher method, suggesting that the drying of MNs progressed further during storage. As shown in Figure 6B, the needle fracture force was inversely correlated with water content in the MNs. The fracture force of MNs reached a plateau after 1 day; hence, the water content in our PVA-based MNs should be maintained at less than 1.59%. McCrudden et al. [42] reported that the water content of MNs fabricated from poly (methylvinyl ether/maleic acid) was 7.22%; thus, the appropriate water content depends on the base materials. There have been reports of dissolving MNs being exposed to high humidity conditions [21,43]; however, research on the impact of varying the residual water content during the manufacturing process is limited. Water content can play a major role in ensuring the optimum mechanical strength of MNs. To this end, we only used PVA-based MNs that had been stored for at least 2 days, thus controlling the effect of water content on mechanical strength during all trials in this study.

Unlike other types of MNs, dissolving MNs are fabricated by encapsulating the active pharmaceutical ingredients (APIs) into the base material(s). We determined the relationship between the amount of API loaded into the MNs and their mechanical strength. We used LID as a model API because it is one of the most widely loaded drugs in dissolving MNs, and its concentration-dependent effects on MN mechanical strength have not been determined [44,45]. The concentration of LID in PVA aqueous solution was 3.75, 7.5, and 15% (*w*/*w*), and the drug content in needles was 38.3 ± 3.5, 86.8 ± 3.1, and 157.0 ± 5.0 µg/patch, respectively. As shown in Figure 7, the needle fracture force gradually decreased with an increase in LID loading amount. The needle fracture force significantly decreased by 39.5% on the incorporation of 15% LID (*w*/*w*) as compared to that of the MNs without LID (one-way ANOVA, *p* < 0.05). A previous study demonstrated that encapsulation of 2–10% calcein in PLGA-based MNs weakened needle fracture force [19]. This may have been due to calcein particles being mechanically weaker than PLGA and the poor adhesion between calcein particles and the PLGA matrix, which provided sites for mechanical failure. We presume the possibility of similar phenomena in our case. Our results show that it is necessary to determine the influence of drugs on needle fracture force and to wisely choose appropriate drug content without affecting needle mechanical strength.

## 4. Discussion

Dissolving MNs have the potential for widespread clinical use, especially in developing countries that lack adequate public health resources. One of the major challenges in MN commercialization is the lack of consensus on the evaluation of this novel dosage form. Due to the diversity of needle properties, including shape and array densities, it is difficult to adopt an evaluation method for conventional injections for dissolving MNs. In this study, we highlighted the importance of mechanical characterization test protocols of this delivery system.

Our findings demonstrate that the mechanical characterization test results are strongly affected by the compression speed and number of compression needles. Richeton et al. showed that the mechanical response of an amorphous polymer is strongly affected by the strain speed [46]. An increase in the compression speed results in an increase in the strain speed. Dissolving MNs are made of an amorphous polymer, which is why the needle fracture force may increase with an increase in the compression speed. The effect of needle numbers could be attributed to the small differences in the timing of individual needle buckling. We used a microscope to observe the needle buckling behavior on video and found a slight time difference between contact with the needle tips and when the needles bent. It is presumed that the peaks on the force–displacement curve were probably averaged out due to the stronger effect of this phenomenon when the number of needles was increased.

The needle fracture force is not an absolute value but a value dependent on test conditions. Hence, the results need to be interpreted with caution. In general, the needle fracture force is used to calculate a margin of safety over the force required for insertion into the skin. Park et al. [18] reported that the insertion force required for MNs with a tip diameter of 25 µm to puncture skin was 0.058 N/needle. Most researchers compared the needle fracture force of MNs with the insertion force reported by Park et al. due to the difficulty of measuring the actual insertion force using animal or human skin [47,48]. Our findings demonstrate that this approach has some limitations and risks associated with ensuring the mechanical strength of dissolving MNs.

To ensure puncture performance, it is practical to control the performance and quality of the applicator and the mechanical strength of dissolving MNs. It is necessary to justify their specifications that should be within appropriate limits, ranges, and distributions. To compare the needle fracture force of dissolving MNs, it is essential to set the same compression speed and number of compression needles. Du et al. [39,49] reported a method to assess the needle fracture force of individual needles using a micromanipulation technique. This method is an effective tool for not only characterizing the variations among single needles in the patch but also measuring the needle fracture force with the same test condition. Combining the micromanipulation technique with our method would provide valuable information for comparing MNs.

There are no reports on the evaluation of dissolving MNs with a wide range of aspect ratios. Using specific conditions, our investigation has revealed that dissolving MNs can be divided into two groups based on their aspect ratio, exhibiting either buckling or unbuckling deformation. Dissolving MNs that have a high aspect ratio required that their needle fracture force be evaluated in detail. Donnelly et al. [25,29] determined the mechanical strength of dissolving MNs by evaluating the shortening of needle height when a constant compression force was applied. This method seems difficult for evaluating dissolving MNs that exhibit buckling deformation. We clearly found that in dissolving MNs, factors such as the needle aspect ratio, water content, and drug content affect needle fracture force and should be given serious consideration as important parameters. It is essential to meet the specification limits for water content before the shipment of each batch. Furthermore, it is vital to evaluate the hygroscopicity of dissolving MNs and define a period in which the mechanical strength can be maintained after opening the package. Although further investigations are needed for different kinds of base polymers and drugs, there might be threshold values for drug content that decrease the mechanical strength of dissolving MNs. Modeling and simulation approaches utilizing these input parameters may be useful for predicting needle fracture force. Future studies are required for the construction of a model to optimize the design of MNs.

Clinically, dissolving MNs would be applied to the skin using an applicator to improve skin puncturability [50,51]. Recently, we observed that needles with a small base diameter broke near the base without bending tips during skin insertion, which was confirmed by micro-CT imaging. This may be attributed to the transverse force applied to the MNs due to the viscoelasticity of the skin. During the non-clinical stage, apart from measuring needle fracture force, skin puncturability should also be evaluated. Although our study provides insightful information, we believe that joint research with the industry, government, and academia for inter-laboratory reproducibility of investigated mechanical characterization tests is necessary for future studies. The findings of our study can aid in exploring the appropriate parameters for the experimental setup and optimization of dissolving MNs.

## 5. Conclusions

This is the first report detailing the influence of the test conditions on the mechanical characterization of dissolving MNs. Our findings showed that the compression speed and number of compression needles significantly affected the needle fracture force. Owing to the selected test conditions, dissolving MNs were categorized into two groups displaying buckling or unbuckling deformation, which relies on the needle aspect ratio. Our findings also showed that the needle aspect ratio, water content, and drug content affected the needle fracture force of these systems. Appropriate parameters for experimental setup are important to characterize the mechanical properties of dissolving MNs for quality evaluation.

## Figures and Tables

**Figure 1 pharmaceutics-16-00200-f001:**
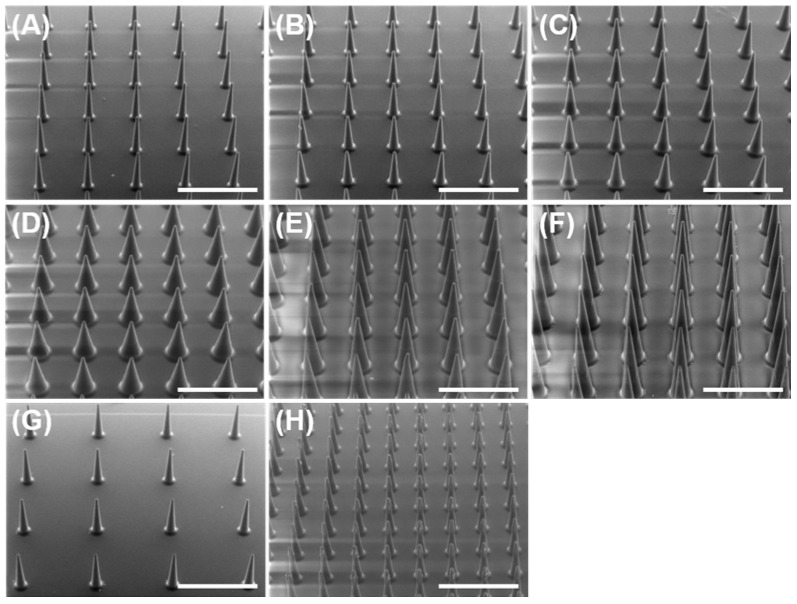
Scanning electron microscopic images of PVA-based dissolving microneedles: (**A**) Model 1, (**B**) Model 2, (**C**) Model 3, (**D**) Model 4, (**E**) Model 5, (**F**) Model 6, (**G**) Model 7, and (**H**) Model 8. Scale bar represents 1000 µm.

**Figure 2 pharmaceutics-16-00200-f002:**
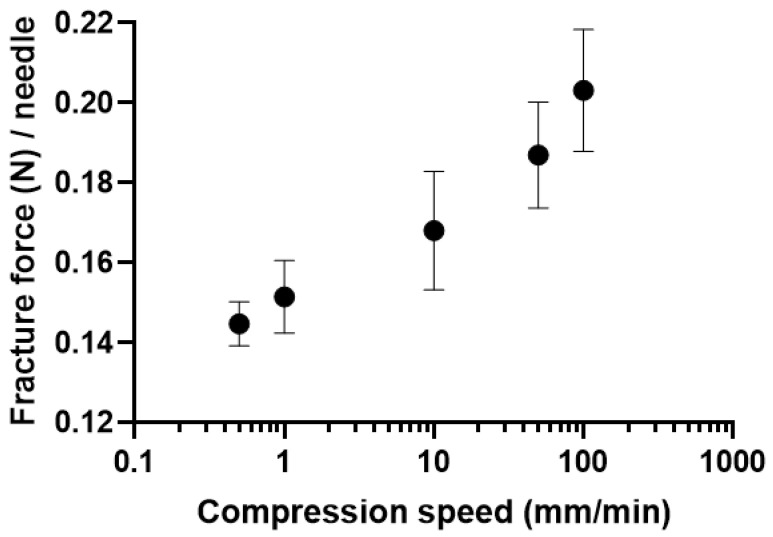
Effect of compression speed on the mechanical fracture force of dissolving microneedles (MNs). The needle fracture force of PVA-based Model 2 MNs was evaluated as a function of compression speed using a 5 mm diameter rod. Results are expressed as mean ± SD (*n* = 4).

**Figure 3 pharmaceutics-16-00200-f003:**
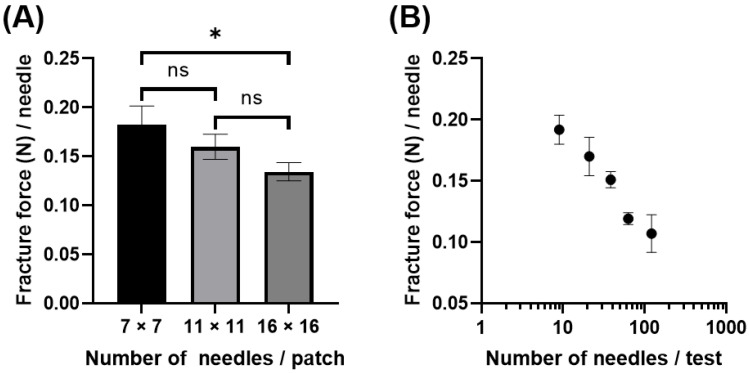
Effect of number of needles during the test on mechanical fracture force of dissolving microneedles (MNs). (**A**) The needle fracture force of PVA-based MNs (Models 7, 2, and 8) having 7 × 7, 11 × 11, and 16 × 16 needles per patch, respectively, using a 5 mm diameter rod. (**B**) The needle fracture force of PVA-based Model 2 MNs that have 121 (11 × 11) needles per patch using different diameters of rods. The tests were conducted at a compression speed of 0.5 mm/min. An asterisk indicates a significant difference in the needle fracture force (one-way ANOVA, *p* < 0.05). Results are expressed as mean ± SD (*n* = 4). ns: not significant.

**Figure 4 pharmaceutics-16-00200-f004:**
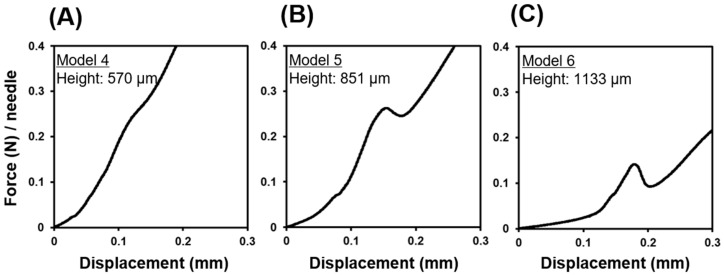
Force–displacement curve of dissolving microneedles (MNs) in terms of needle aspect ratio. Representative force–displacement curve of PVA-based MNs, (**A**) Model 4, (**B**) Model 5, and (**C**) Model 6, with actual needle heights of 570, 851, and 1133 µm, respectively, and a fixed design base diameter of 300 µm. The tests were conducted at a compression speed of 0.5 mm/min using a 5 mm diameter rod.

**Figure 5 pharmaceutics-16-00200-f005:**
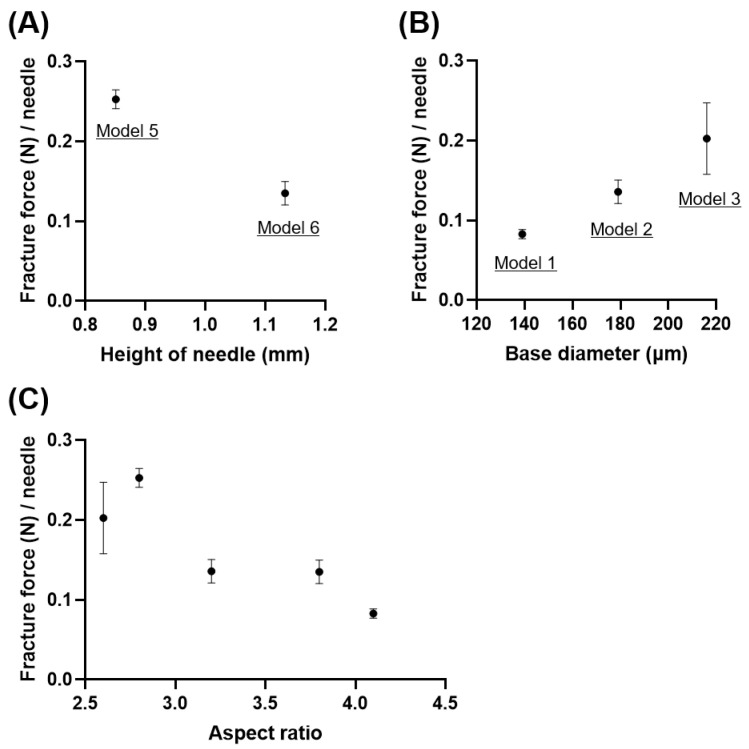
Mechanical fracture force analysis of dissolving microneedles (MNs) in terms of the needle aspect ratio. (**A**) Needle fracture force of PVA-based MNs as a function of actual needle height with a fixed design base diameter of 300 µm. (**B**) Needle fracture force of PVA-based MNs as a function of base diameter with a fixed design needle height of 600 µm. (**C**) Needle fracture force of PVA-based MNs as a function of the aspect ratio (needle height/base diameter). The tests were conducted at a compression speed of 0.5 mm/min using a 5 mm diameter rod. Results are expressed as mean ± SD (*n* = 3–4).

**Figure 6 pharmaceutics-16-00200-f006:**
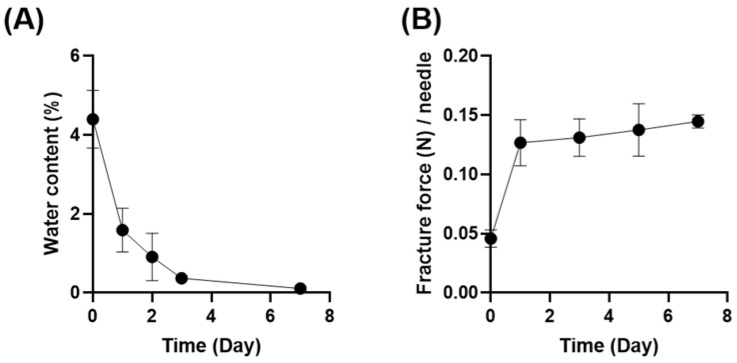
Mechanical fracture force analysis of dissolving microneedles (MNs) in terms of residual water content. (**A**) Water content and (**B**) needle fracture force of PVA-based Model 2 MNs as a function of storage period. The tests were conducted at a compression speed of 0.5 mm/min using a 5 mm diameter rod. Results are expressed as mean ± SD (*n* = 3–4).

**Figure 7 pharmaceutics-16-00200-f007:**
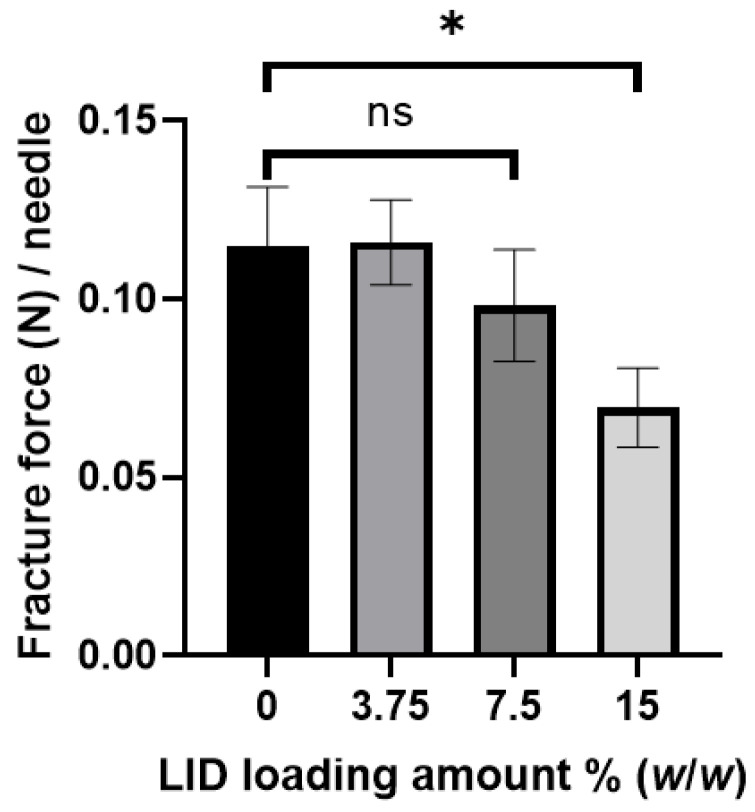
Mechanical fracture force analysis of dissolving microneedles (MNs) in terms of drug content. The needle fracture force of lidocaine hydrochloride (LID)-loaded PVA-based Model 2 MNs as a function of LID loading content. The tests were conducted at a compression speed of 0.5 mm/min using a 5 mm diameter rod. An asterisk indicates a significant difference in the needle fracture force (one-way ANOVA, *p* < 0.05). Results are expressed as mean ± SD (*n* = 4). ns: not significant.

**Table 1 pharmaceutics-16-00200-t001:** Microneedle (MN) master mold design dimensions for micromolding technology.

Model No.	Height (µm)	Base Diameter (µm)	Aspect Ratio	Interspacing of MNs at Tip (µm)	MN Density (/Patch)
1	600	120	5	600	121 (11 × 11)
2	600	150	4	600	121 (11 × 11)
3	600	200	3	600	121 (11 × 11)
4	600	300	2	600	121 (11 × 11)
5	900	300	3	600	121 (11 × 11)
6	1200	300	4	600	121 (11 × 11)
7	600	150	4	1000	49 (7 × 7)
8	600	150	4	400	256 (16 × 16)

The MN shape and tip diameter were conical and 40 µm, respectively.

**Table 2 pharmaceutics-16-00200-t002:** The actual dimensions of PVA-based microneedles (MNs) fabricated using micromolding technology.

Model No.	Height (µm)	Base Diameter (µm)	Aspect Ratio	Interspacing of MNs at Tip (µm)	Tip Diameter (µm)
1	574 ± 4	139 ± 6	4.1 ± 0.2	575 ± 4	29 ± 1
2	573 ± 2	179 ± 4	3.2 ± 0.1	578 ± 3	31 ± 1
3	571 ± 1	216 ± 4	2.6 ± 0.0	573 ± 3	35 ± 1
4	570 ± 2	310 ± 3	1.8 ± 0.0	574 ± 2	29 ± 1
5	851 ± 3	299 ± 4	2.8 ± 0.0	576 ± 4	32 ± 1
6	1133 ± 3	300 ± 3	3.8 ± 0.0	581 ± 2	31 ± 1
7	575 ± 3	172 ± 5	3.3 ± 0.1	968 ± 4	31 ± 1
8	574 ± 2	174 ± 4	3.3 ± 0.1	386 ± 3	30 ± 1

Mean ± S.D., *n* = 18.

## Data Availability

Data are contained within the article.

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
