# Peer review of "Mechanical Characterization of Dissolving Microneedles: Factors Affecting Physical Strength of Needles"

_pharmaceutics, 2024, doi:10.3390/pharmaceutics16020200_

Round 1
Reviewer 1 Report
Comments and Suggestions for Authors
Review of the manuscript pharmaceutics-2787107
Mechanical characterization of dissolving microneedles: factors affecting physical strength of needles
The authors have provided experimental evidence of the selection of several appropriate parameters to characterize the production of dissolving microneedles. This is an essential aspect that can help transform an "academic" device into a medical one. Overall, this is a good paper, the design of the experiment is good, the methods are adequately described, and the results are convincing. The structure of the paper can be improved, by adding a specific discussion section and moving some discussion present in the results section. To further improve the quality of the manuscript, here is a list of comments/recommendations to the authors:
Line 183: high aspect ratio (800 µm in height): the value of the AR could be added, or the base diameter.
Lines 230-232: not clear, the authors could add a comment about the buckling effect itself, for readers that are not familiar with it. Again, any reference in the literature that supports this finding or analysis may help (as the authors did in Line 270 about the correlation between the needle’s AR and fracture force). I recommend moving this discussion after line 325.
Fig 3B: please check that the fourth point from the left has no (or very little) error bar (same comment for Fig5B)
Discussion of Fig. 4: it is probably not easy to estimate the buckling limit of a conical needle, but is there any simplified model based on Euler’s formula that can be provided to refine the analysis and provide design rules?
Fig. 5: I recommend adding a plot fracture force versus AR to support the discussion line 275.
Section 3.5: as suggested by this result, I let the authors double-check if this parameter (water content) is identified and under control during the different trials made in the study.
Line 325: this section (discussion) appears suddenly, without a title, please revise.
Fig 2: the authors could discuss further this graph and suggest some mechanisms to explain the trend or find indications in previous research that support this finding.
I leave it to the authors to consider the possibility of adding a summary table or diagram to summarize the results.
Reviewer 2 Report
Comments and Suggestions for Authors
The manuscript contains the mechanical characterization of dissolving microneedles. This manuscript demonstrates the differences in needle breaking force of dissolving microneedles prepared under different conditions in terms of needle aspect ratio (needle height/base diameter), moisture content, and drug content in a simple microneedle structure. To be published in this journal, new content must be covered compared to many other journals on drug-laden microneedles. Therefore, this manuscript is not suitable for publication.
Comments on the Quality of English LanguageNo problem.
Reviewer 3 Report
Comments and Suggestions for Authors
Abstract:
The author should provide in the abstract the rationale for the selection of polyvinyl alcohol (PVA) as the base material for the dissolving microneedle patches.
Could the authors provide some details about the observed buckling and unbuckling deformations in PVA-based microneedles? How do these deformations correlate with the aspect ratio?
The authors should mention some results of needle fracture force under various conditions, including needle aspect ratio, water content, and drug content.
In the background of transdermal drug delivery, how do you predict the findings of this study will contribute to the design and development of dissolving microneedles for specific drug delivery applications?
Introduction:
Can the authors elaborate more on the advantages and limitations of each of the four categories of microneedles mentioned in the study?
What are the major challenges in gaining approval for dissolving microneedles for medical use from regulatory bodies such as the U.S. Food and Drug Administration and the Pharmaceuticals and Medical Devices Agency of Japan?
Can the authors explain the significance of mechanical strength in dissolving microneedles and how it differs from conventional solid microneedles?
Methods
How were the reproducibility and consistency of the microneedles ensured, and can you discuss more on the rationale behind the chosen storage conditions in aluminum plastic-laminated packaging with dry silica gel at 25 °C?
How did the choice of the diameters (2, 3, 4, 5, or 13.3 mm) contribute to the complete evaluation of the mechanical strength of the microneedles?
Could the authors mention why dissolving microneedles were stored for over 2 days?
Could the authors provide more details on the precision and sensitivity of Karl Fischer titration for measuring water content in the MN patches?
Results and discussion
Are there any specific factors that might explain the observed difference in penetration effectiveness based on MN height? “PVA-based dissolving MNs higher than 600 μm could puncture rat skin more effectively than those with a height of 300 μm”.
Are there any potential implications of the shrinkage on the performance of the microneedles that should be considered?
The findings of needle fracture force that influence the practical application of dissolving microneedles need more elaboration.
How might the controlling water content be important for ensuring the mechanical integrity of dissolving microneedles?
The authors should provide more discussion on the residual water content and drug loading, the observed trends, and the implications of these properties on the performance of microneedles for drug delivery?
Comments on the Quality of English Language
A minor edition is required.
Round 2
Reviewer 1 Report
Comments and Suggestions for Authors I want to thank the authors for conscientiously considering all the comments made during the revision of the manuscript. I'll leave it to the authors to check on line 79 whether the term "silicone" - the elastomer material - is correct, or whether silicon (Si) could be used instead. I recommend publishing the revised manuscript in Pharmaceutics.